# Transcranial Magnetic Stimulation to Treat Neuropathic Pain: A Bibliometric Analysis

**DOI:** 10.3390/healthcare12050555

**Published:** 2024-02-28

**Authors:** Bruno Daniel Carneiro, Isaura Tavares

**Affiliations:** 1Unit of Experimental Biology, Department of Biomedicine, Faculty of Medicine, University of Porto, 4200-319 Porto, Portugal; isatav@med.up.pt; 2Institute for Research and Innovation in Health and IBMC, University of Porto, 4200-135 Porto, Portugal

**Keywords:** pain, neuropathic pain, transcranial magnetic stimulation, treatment, bibliometric analysis

## Abstract

Neuropathic pain is caused by a lesion or disease of the somatosensory system and is one of the most incapacitating pain types, representing a significant non-met medical need. Due to the increase in research in the field and since innovative therapeutic strategies are required, namely in intractable neuropathic pain, neurostimulation has been used. Within this approach, transcranial magnetic stimulation (TMS) that uses a transient magnetic field to produce electrical currents over the cortex emerges as a popular method in the literature. Since this is an area in expansion and due to the putative role of TMS, we performed a bibliometric analysis in Scopus with the primary objective of identifying the scientific production related to the use of TMS to manage neuropathic pain. The research had no restrictions, and the analysis focused on the characteristics of the literature retrieved, scientific collaboration and main research topics from inception to 6 July 2023. A total of 474 articles were collected. A biggest co-occurrence between the terms “neuropathic pain” and “transcranial magnetic stimulation” was obtained. The journal “Clinical Neurophysiology” leads the Top 5 most productive sources. The United States is the most productive country, with 50% of US documents being “review articles”, followed by France, with 56% of French documents being “original articles”. Lefaucheur, JP and Saitoh, Y are the two most influential authors. The most frequent type of document was “original article”. Most of the studies (34%) that identified the neuropathic pain type focused on traumatic neuropathic pain, although a large proportion (38%) did not report the neuropathic pain type. This study allows us to provide a general overview of the field of TMS application for neuropathic pain and is useful for establishing future directions of research in this field.

## 1. Introduction

Neuropathic pain is one of the most incapacitating pain types representing a significant non-met medical need [1]. There are several conditions that can cause neuropathic pain such as traumatic and metabolic nerve lesions, infections, discal disease, multiple sclerosis, spinal cord injury, head trauma or stroke [1].

There have been several challenges in the definition of neuropathic pain. Neuropathic pain was defined as “pain initiated or caused by a primary lesion or dysfunction of the nervous system” [2] in 1994, and in 2008, the definition was modified to “pain arising as a direct consequence of a lesion or disease affecting the somatosensory system” [2]. In 2011, a novel definition of neuropathic pain emerged as “pain caused by a lesion or disease of the somatosensory system” [3]. Absent from the definition is the term “dysfunction” because some chronic pain conditions, such as fibromyalgia or complex regional pain syndrome, are central nervous system-related dysfunctions without evidence of nerve lesion or injury in the tissue. The International Association for the Study of Pain (IASP) proposed, in 2017, a new classification for chronic pain [4] that includes neuropathic pain [5], and this condition is nowadays integrated into the International Classification of Diseases and Related Health Problems (ICD-11) of the World Health Organization (WHO) [6]. The neuropathic pain classification differentiates pain of peripheral origin and pain of central origin and encompasses nine conditions, such as postherpetic neuralgia, painful nerve lesion and painful neuropathies [5]. Each condition is part of detailed models that describe investigations supporting a definitive diagnosis and contain codes for temporal aspects, psychosocial factors and severity associated with pain. Interestingly, there is no requirement for chronicity to meet the current definition of neuropathic pain. However, chronicity is often assumed. In fact, the update of the definition wants to show that this pain may occur acutely [3]. Post-surgical pain, acute sciatica and Guillain–Barré syndrome are some examples of conditions that can induce acute neuropathic pain.

The management of neuropathic pain remains a real challenge despite some promising results of drugs acting on novel targets [7,8]. In fact, some recent meta-analysis evaluations highlighted that about 30% to 40% of patients with neuropathic pain have an appropriate response when compared with placebo and the efficacy of drugs has become even poorer [9,10]. Despite some exciting results with several targets and classes of drugs, different therapeutic strategies are required, namely for treating intractable neuropathic pain. These strategies include rational combination therapy; drug repositioning; individualized pain management, including cognitive-behavioral approaches and neurostimulation techniques.

Included in the group of the neurostimulation techniques, a non-invasive brain stimulation technique emerged in the last few decades, namely transcranial magnetic stimulation (TMS). TMS uses a transient magnetic field to produce electrical currents over the cortex [11], and neuropathic pain seems to be shot via high-frequency (about 5 Hz to 20 Hz) stimulation of the contralateral primary motor cortex and using stimulation of high intensity (about 1500 to 2000 pulses per session) [12,13]. Two recent studies with placebo control using robotized neuronavigation and a double-blind design confirmed the efficacy of sessions of TMS on the primary motor cortex in patients with peripheral [14] and central neuropathic pain [15]. Safety was excellent, with transient headache being the main side-effect. According to Lefaucheur et al., level A evidence is proposed for the use of high-frequency TMS over the primary motor cortex contralateral to the pain side in the management of neuropathic pain [11]. Level A evidence means definite efficacy [11].

Based on the challenges involved in neuropathic pain management and the putative role of TMS, we performed an analysis of the literature using a bibliometric approach. This approach has been used recently to provide a general overview of the trend of publication in a certain field of research and has considerable advantages over narrative reviews, namely its higher objectivity and intuitive analysis [16]. Interestingly, a group of authors recently carried out a bibliometric analysis that aimed to provide a bibliometric perspective regarding articles on pain and transcranial direct current stimulation (TDCS) [17]. TDCS is different to TMS because TDCS works by using a low-voltage source of electricity that delivers a fixed current of low intensity between two electrodes placed on the scalp of the patient [18]. Despite these different approaches, that study, along with the present one, shows the utility of bibliometric analysis. To the best of our knowledge, however, there is no bibliometric analysis that focuses specifically on the role of TMS in treating neuropathic pain, so it is relevant to perform such a study. The primary objective of this study was to identify the scientific production linked to the use of TMS to treat neuropathic pain and provide an overview of the research developments in this field.

## 2. Materials and Methods

### 2.1. Study Design

A bibliometric analysis was carried out to evaluate the research on the use of TMS to treat neuropathic pain from inception to 6 July 2023. The search had no restrictions.

### 2.2. Bibliographic Database, Journal Impact Factor Value Source and Field-Weighted Citation Impact

The bibliographic database used in this study took into consideration the broadest possible coverage regarding publications that address the subject under study. We opted for Scopus, a peer-reviewed database that is a generalized and broader database that allows us to use several tools for data extraction and aggregation in different formats, providing detailed information for the analysis. Furthermore, Scopus is the largest abstract and citation database. Another reason to choose Scopus over other databases was that it provides bibliometric information previously organized and systematized, which makes it easier to collect the data and perform the analysis. The value of each Journal Impact Factor (JIF) was obtained from the Journal Citation Reports and indicates the measure of the frequency with which the average article in a journal has been cited in a particular year. The Field-Weighted Citation Impact (FWCI) indicates the number of citations received by an article compared to the average or expected number of citations received by similar publications.

### 2.3. Search Expression

Firstly, we defined a search term using terms found in relevant articles or documents, and then we calibrated the search [19] through testing attempts, considering combined and separate terms. The final search expression was obtained with the TITLE-ABS-KEY filter as follows: [(“neuropathic pain”) AND (“transcranial magnetic stimulation” OR “repetitive transcranial magnetic stimulation” OR “deep transcranial magnetic stimulation” OR “single-pulse transcranial magnetic stimulation” OR “rapid rate transcranial magnet stimulation” OR “transcranial magnetic stimulation modality” OR “TMS” OR “rTMS” OR “dTMS” OR “noninvasive brain stimulation”) AND (“treatment” OR “management” OR “treat” OR “therapy” OR “therapeutic”)].

### 2.4. Software and Data Analysis

All articles were analyzed by index and author keywords, source, publication year, type of article, country, top-cited articles, authorship and by type of neuropathic pain. Index keywords and author keywords were analyzed by frequency. We performed a ranking of publication sources by relevance. The total and the average number of citations were analyzed by country. By computing the number of multi-authored and single publications, we performed the co-authorship and authorship analyses. In the analyses, we used the VOSviewer [20] version 1.6.19, a software program specifically designed for this type of analysis. VOSviewer allows for the visualization of similarities and allows us to build a similarity matrix from a co-occurrence matrix using association strength [21]. The result is a two-dimensional map where similarity is reflected by the distance between elements [21]. Microsoft Excel version 16.82 supported the preparation of the data exported from Scopus to perform the analysis. To mitigate possible errors, we analyzed each document in order to remove duplicates and erroneous entries.

## 3. Results

A search of Scopus was performed on 6 July 2023. A total of 474 documents were obtained.

### 3.1. Keywords

The minimum number of occurrences of a keyword was set at 25. Of 4470 keywords, 119 met this minimum threshold (25 occurrences). For each of 119 keywords, the total strengths of the co-occurrence links with other keywords were calculated, and the keywords with the highest total link strengths were selected. More general terms such as “human” (423 occurrences), “humans” (312 occurrences), “article” (199 occurrences), “review” (183 occurrences), “female” (155 occurrences) or “male” (152 occurrences) were not considered for this analysis.

The Top 10 most relevant index or author keywords and their frequencies were “neuropathic pain” (401 occurrences; total link strength of 3894), “transcranial magnetic stimulation” (377 occurrences; total link strength of 3707), “analgesia” (173 occurrences; total link strength of 2075), “repetitive transcranial magnetic stimulation” (166 occurrences; total link strength of 1783), “motor cortex” (164 occurrences; total link strength of 1770), “neuralgia” (159 occurrences; total link strength of 1720), “chronic pain” (153 occurrences; total link strength of 1649), “brain depth stimulation” (139 occurrences; total link strength of 1580), “pain” (125 occurrences; total link strength of 1330) and “treatment outcome” (108 occurrences; total link strength of 1205).

The co-occurrence of terms and words related to this study shows that the chosen search expression was relevant. This analysis was performed using the VOSviewer, which allowed us to construct Figure 1. The highest co-occurrence appears between “neuropathic pain” and “transcranial magnetic stimulation”.

### 3.2. Publication Year

The results in Figure 2 show that interest in the use of TMS to treat neuropathic pain began at the start of the twenty-first century with the first publication about this theme in 2001 [22]. More precisely, the first results in this domain were reported at the First International Symposium on Transcranial Magnetic Stimulation, held in Göttingen, Germany, in October 1998, and published as an abstract in *Clinical Neurophysiology* [23]. By the following year, more publications in this area begun to appear, and in 2013, this topic started to appear more regularly in the literature. In the past few years, namely 2020, 2021 and 2022, this topic has obtained even more attention from investigators, with 2022 being the year with the most publications (n = 48). The largest increase in publications happened between 2010 and 2011 (from 12 to 26 documents). Figure 2 does not include the year 2023, because the data from that year only cover half of the year and could mislead the reader to believe a major decrease in the number of publications occurred during this year; however, it should be noted that until 6 July 2023, there had been a total of 13 publications.

### 3.3. Source

Figure 3 shows the number of publications by the five most productive sources over the years. The source with the highest number of publications was the journal “Clinical Neurophysiology” (2022 JIF of 4.7), with a total of 25 publications (n = 25) since 2004. The Top 10 most productive sources also included “Brain Stimulation” (2022 JIF of 7.7; n = 16), “Pain” (2022 JIF of 7.4; n = 14), “Neuromodulation” (2022 JIF of 2.8; n = 13), “Current Pain and Headache Reports” (2022 JIF of 3.7; n = 11), “Brain Sciences” (2022 JIF of 3.3; n = 8), “European Journal of Pain” (2022 JIF of 3.6; n = 8), “Pain Physician” (2022 JIF of 3.7; n = 8), “Journal of Pain” (2022 JIF of 4.0; n = 7) and “Frontiers in Human Neuroscience” (2022 JIF of 2.9; n = 6). In 2022, the source of the Top 10 most productive sources with the most publications was “Neuromodulation”, with a total of four documents.

### 3.4. Study Type

As shown in Figure 4, the three most frequent types of documents in the results were “original article” (n = 219, 46.2%), “review article” (n = 187, 39.5%) and “book chapter” (n = 18, 3.8%). Other publication types include “editorial” (n = 16), “conference paper” (n = 14), “letter” (n = 8), “short survey” (n = 7), “note” (n = 4) and “book” (n = 1). Regarding the “original article” type, two periods had the highest number of publications: from 2013 to 2016 (n = 62) and 2018 to 2021 (n = 79). Moreover, the highest number of publications of the “review article” type occurred in two periods: from 2014 to 2017 (n = 41) and 2020 to 2022 (n = 60). The year with the higher number of publications was 2019 (n = 20) for the “original article” type and 2022 (n = 27) for the “review article” type.

### 3.5. Country

The country assessment took into account the country of affiliation of the first author of the articles as the unit of analysis. Notably, as shown in Figure 5, most of the documents originated from the United States (n = 90, 19.0%), which had the highest number of publications in 2014 (n = 14). Top 10 most productive countries also included France (n = 80, 16.9%), China (n = 44, 9.3%), the United Kingdom (n = 29, 6.1%), Germany (n = 28, 5.9%), Italy (n = 27, 5.7%), Japan (n = 26, 5.5%), Brazil (n = 19, 4%), Canada (n = 15, 3.2%) and South Korea (n = 12, 2.5%). Interestingly, in the United States around, 50.0% of the documents were “review articles” (n = 45) and around 34.4% of the documents were “original articles” (n = 31), and in France, around 56.3% of the documents were “original articles” (n = 45) and around 23.8% of the documents were “review articles” (n = 19).

### 3.6. Authors and Co-Authorship

The authors with the most considerable number of publications were Lefaucheur, JP (n = 48), Saitoh, Y (n = 23) and Garcia-Larrea, L (n = 16). In the Top 10 list of authors in this field, we can also see Fregni, F and Nguyen, JP (both with n = 13), Ayache, SS and Bouhassira, D (both with n = 12), André-Obadia, N and Hosomi, K (both with n = 10), and Attal, N (n = 9).

In the co-authorship evaluation, we used VOSviewer to analyze the association strength, and a maximum number of 25 authors per document and a minimum number of 4 documents per author were considered. As shown in Figure 6, nine co-authorship interconnected clusters were obtained from the analysis. The authors with the highest strength link were Lefaucheur, JP and Saitoh, Y.

### 3.7. Most-Cited Publications

It is important to say that some documents in the first places of the Top 10 most-cited publications lists are not directly related to the primary subject of this article (transcranial magnetic stimulation to treat neuropathic pain). Due to that, we highlighted here only the three most-cited “original articles” and the three most-cited “review articles” directly related to the primary subject of this study.

The three most-cited “original articles” were as follows: in first place, “Longlasting antalgic effects of daily sessions of repetitive transcranial magnetic stimulation in central and peripheral neuropathic pain” [24], published on 2005 by the “Journal of Neurology, Neurosurgery and Psychiatry”, with a total of 350 citations so far and a 2023 FWCI of 7.64; in second place, “Motor cortex rTMS restores defective intracortical inhibition in chronic neuropathic pain” [25], published on 2006 by “Neurology”, with a total of 309 citations and a 2023 FWCI of 3.13; in third place, “Transcranial magnetic stimulation for pain control. Double-blind study of different frequencies against placebo, and correlation with motor cortex stimulation efficacy” [26], published on 2006 by “Clinical Neurophysiology”, with a 2023 FWCI of 3.76 and a total of 208 citations.

The three most-cited “review articles” were as follows: in first place, “Evidence-based guidelines on the therapeutic use of repetitive transcranial magnetic stimulation (rTMS)” [27], published on 2014 by “Clinical Neurophysiology”, with 1396 citations and a 2023 FWCI of 20.81; in second place, “Evidence-based guidelines on the therapeutic use of repetitive transcranial magnetic stimulation (rTMS): An update (2014–2018)” [11], published on 2020 by “Clinical Neurophysiology”, with 767 citations and a 2023 FWCI of 33.42; in third place, “rTMS for Suppressing Neuropathic Pain: A Meta-Analysis” [28], published on 2009 by “Journal of Pain”, with a 2023 FWCI of 2.46 and a total of 178 citations so far.

### 3.8. Type of Neuropathic Pain

Regarding the type of neuropathic pain addressed in the publications, a large number of studies (38.0%; n = 180) do not report the type of neuropathic pain treated by TMS and address neuropathic pain in general. Around 33.8% (n = 160) of the publications focus on traumatic neuropathic pain. Of the remaining publications (28.2%; n = 134), 57 publications (12.0%) focus on neuropathic pain related with neurological diseases (for example, Parkinson’s disease or multiple sclerosis), 25 publications (5.2%) on neuropathic pain related to cardiovascular diseases (namely stroke), 13 publications (2.8%) on neuropathic pain in the context of phantom limb pain, 9 publications (1.9%) on neuropathic pain associated with cancer and 8 publications (1.7%) on neuropathic pain associated with fibromyalgia. Very few publications address diabetic neuropathic pain (n = 3), infections (namely postherpetic neuralgia, n = 2), regional pain syndrome (n = 7) and burning mouth syndrome (n = 2).

## 4. Discussion

The presented study analyzed a total of 474 publications from 22 years (from the inception of research in the field and until 6 July 2023) about the use of TMS to treat neuropathic pain. To the best of our knowledge, this was the first bibliometric analysis that focused on the existing research on the use of TMS in neuropathic pain. It seems clear the increase in literature on this subject occurred in the past few years (2020, 2021 and 2022), with 2022 being the year with the most publications. It is important to note that the year 2023 has fewer articles published on this topic than the year 2022 because the bibliometric analysis was conducted until 6 July 2023. The continuous increase in publications highlights the increase in research and interest in the application of TMS for neuropathic pain. Herein, our bibliometric analysis indicates that new data will probably emerge soon about neuropathic pain and TMS. This finding is in line with the need for research addressing nonpharmacological measures to treat neuropathic pain with cost-effective solutions and minimal side effects, which has been emphasized by some authors [29,30].

A large co-occurrence between the terms “neuropathic pain” and “transcranial magnetic stimulation” was detected in our analysis since, from the 474 documents analyzed, 451 used at least one of these terms as keywords. This shows that inside this topic, the relationship between neuropathic pain and TMS is considerably visible. The keywords less mentioned but with important link strengths were “neuralgia” and “treatment outcome”. This probably shows that some investigators devote their work to understanding the impact of using TMS on the treatment of neuropathic pain, evaluating the resulting outcomes and understanding the impact of using TMS in the treatment of different types of neuropathic pain, such as neuralgia [31]. In fact, there are studies that suggest that TMS is a reasonable and well-tolerated add-on treatment in neuropathic pain [32,33,34,35].

Regarding the source of the publications, we listed and characterized the Top 10 most-productive sources in this field. Interestingly, of the total of twelve publications published in 2023, none were published by the five most productive sources up to the moment of analysis (6 July 2023). Instead of that, the source with more publications in 2023 was “Brain Sciences” (sixth place in the Top 10 most productive sources). The journal “Clinical Neurophysiology”, the most productive source, is the journal responsible for the publication of the third [26] most-cited “original article” and the publication of the first [27] and second [11] most-cited “review articles”. It is important to highlight that the number of publications by source is relatively sparse, and the Top 10 most productive sources account for 24.5% of the total publications. This shows that it is difficult to find a dominant source in this topic. In our opinion, the absence of a dominant source may be due to the scope of the topic and the growing interest in performing research in this area, which means that different sources are interested in publishing paper on the topic; however, it could also mean that there is no specific source dedicated to research in this area. The conclusion regarding the most productive countries is different, since the presented Top 10 most productive countries make up to a total of 78.1% of the published publications. The Americas and the European continent are the most productive continents; however, research in this area in other continents is increasing, with a good many contributions from Asian countries such as China or South Korea. From the author and co-authorship analysis, it can be concluded that cooperation between different authors from different countries is a reality in this topic, with Lefaucheur, JP and Saitoh, Y being the two most influential authors in this area. A strong co-authorship association was observed between Lefaucheur, JP and Nguyen, JP and between Saitoh, Y and Shimokawa, T. Regarding the type of neuropathic pain addressed in the publications, the large majority of the analyzed studies (38.0%) do not report a particular type of neuropathic pain. The main neuropathic pain type of the addressed studies is traumatic neuropathic pain, which may be related to the severity of this pain type and the complexity of the treatment. Collectively, the results indicate the need for a better report on the type of neuropathy pain and the need to investigate the value of the application of TMS in the management of different types of neuropathic pain, other than traumatic neuropathic pain.

The continuous growth of publications on the use of TMS to manage neuropathic pain over 22 years highlights the investment in research about the efficacy, benefits and risks of TMS. In fact, most of the articles published in the early years were proof-of-concept studies, based on single sessions, in particular for the purpose of predicting the efficacy of subsequent surgically implanted epidural cortical stimulations [36,37], and the first articles mentioning the use of repeated sessions for a truly therapeutic purpose were published in 2004/2005 with a short-term evaluation for a series of patients [23] or a long-term evaluation for a single case [38]. The bibliometric analysis in this study shows that there are lots of publications on this topic that are “original articles”, but if we add up publications of other types, it can be seen that these other types constitute the majority of publications. Nowadays, all original articles published only relate to protocols using repeated TMS sessions for several weeks with a more or less long-term evaluation. More clinical studies can be carried out to strengthen the evidence for using TMS to treat neuropathic pain. It may also be important to investigate the mechanisms underlying the TMS interventions in neuropathic pain, even using translational approaches. TMS is already recommended for the treatment of resistant psychiatric pathology, namely depression disorder [39] or obsessive-compulsive disorder [40], but the mechanisms behind TMS and neuropathic pain complexity rise specific challenges that are not fully understood [24], and clinical studies may be considered heterogeneous [41]. In fact, a main problem in the use of repetitive TMS for a therapeutic purpose in chronic pain patients is the heterogeneity of the proposed protocols, particularly in terms of the definition of the stimulation target and session rhythm. Furthermore, a precise algorithm must be validated, namely the one proposed by Lefaucheur and Nguyen [42]. Future high-quality studies on this topic are definitely needed, and we hope that this bibliometric analysis can help us to identify the direction of the future research.

Regarding the limitations of this study, we highlighted that other scientific databases that were not investigated might have presented relevant publications within the scope of this analysis. However, since Scopus has a broader coverage, we elected to use this scientific base in a manner similar to other authors [43,44]. We recognize that bibliometric data from Scopus are not produced exclusively for bibliometric analysis and, therefore, can contain imprecisions errors, which are not, however, likely to affect the main results of the current bibliometric analysis. In fact, this bibliometric analysis brings a considerable volume of publications resulting from research with no restrictions and consisting of a useful representation of the most up-to-date knowledge on the subject. Furthermore, the analysis of the results was conducted critically and carefully to avoid errors that could arise from the direct analysis of the results obtained from Scopus.

## 5. Conclusions

According to the results of this bibliometric analysis and its scope, the scientific community should know that the treatment of neuropathic pain may include TMS. The still-weak scientific evidence for the use of TMS in the treatment of neuropathic pain should motivate more research in this area to improve the quality of life of patients.

This study offers a contextual outline that may be useful for leading and proposing new studies with more homogeneous methods and offers some clues about the TMS role in the treatment of neuropathic pain and what and how research has been carried out in this area. The results obtained allow us to make some suggestions for future research, namely adding more scientific databases to find more relevant and current publications, performing large scale meta-analysis to obtain more evidence in this field and promoting the conducting of clinical trials that evaluate the role of TMS in the treatment of neuropathic pain.

## Figures and Tables

**Figure 1 healthcare-12-00555-f001:**
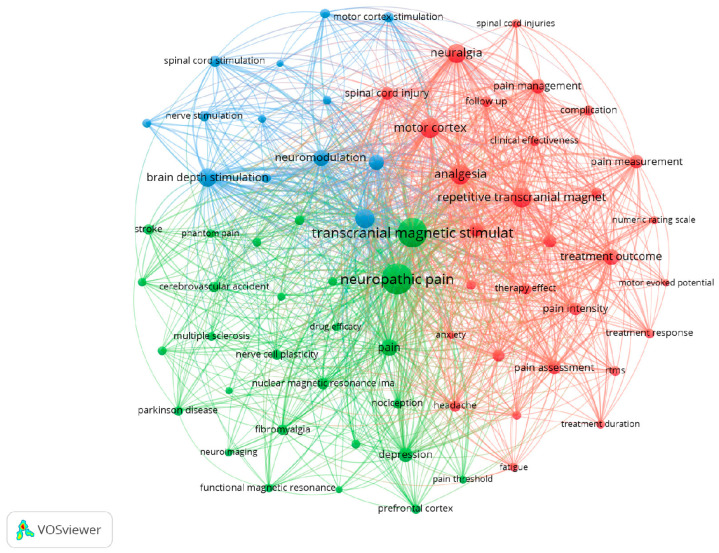
Co-occurrence of author or index keywords. The colors represent the clusters created based on the co-occurrence of keywords. Twenty-five was the minimum number occurrence of words.

**Figure 2 healthcare-12-00555-f002:**
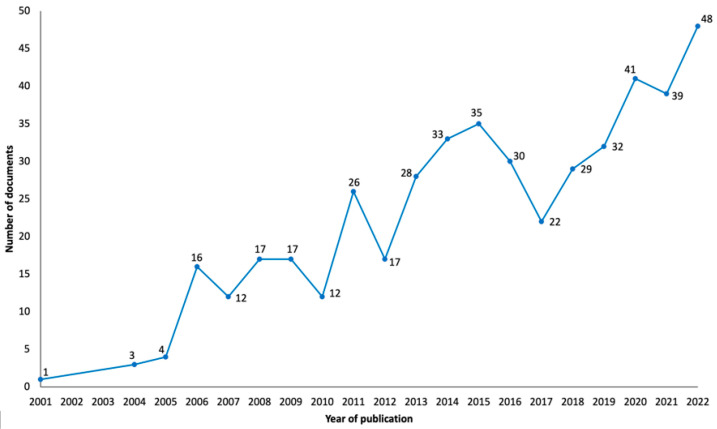
Graphical representation of the evolution of obtained publications since the inception of research into the subject of neuropathic pain and TMS.

**Figure 3 healthcare-12-00555-f003:**
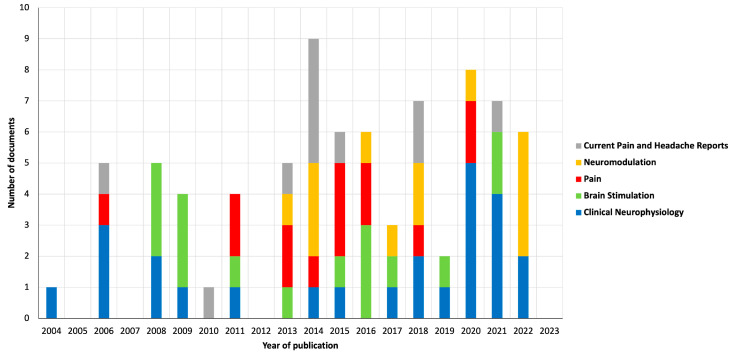
Graphical representation of publications by the five most productive sources.

**Figure 4 healthcare-12-00555-f004:**
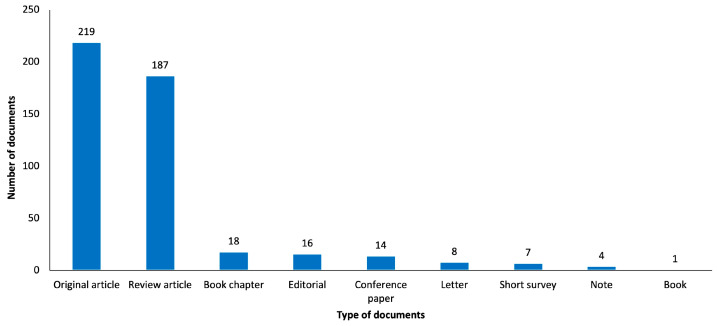
Graphical representation of retrieved publications by type of study.

**Figure 5 healthcare-12-00555-f005:**
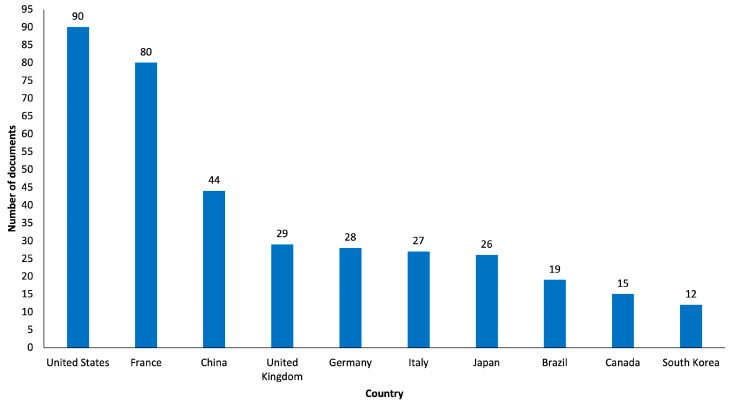
Graphical representation of Top 10 most productive countries.

**Figure 6 healthcare-12-00555-f006:**
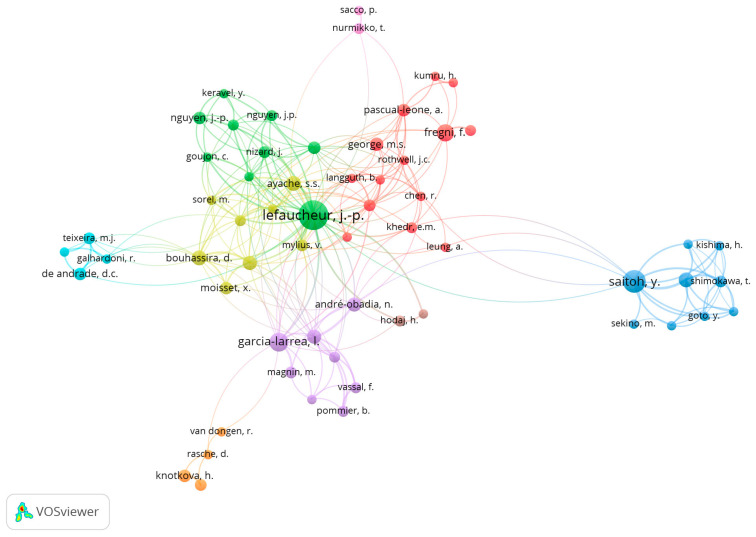
Co-authoring clusters demonstrate the most frequent groups in publications and the relationships with other clusters. The circle’s size is proportional to the document’s number.

## Data Availability

All data are contained within the article.

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
