# Peer review of "Transcranial Magnetic Stimulation to Treat Neuropathic Pain: A Bibliometric Analysis"

_healthcare, 2024, doi:10.3390/healthcare12050555_

Round 1

Reviewer 1 Report

Comments and Suggestions for Authors

The bibliometric analysis of the scientific production linked to the use of TMS to treat neuropathic pain and to provide an overview of the research development is is comprehensive, well thought out and well motivated. There was any shortcomings detected in the manuscript.

The presentation/description of Figure 4 is not very informative and could be improved.

The question for authors to discuss: What is your opinion about the absence of a dominant source in this topic? 

Author Response

We acknowledge the comments of the reviewers. These comments were a remarkable contribute to improve the manuscript. A detailed reply to all the comments was included bellow and the manuscript was changed accordingly, as detailed bellow.

Reviewer 1

1. The bibliometric analysis of the scientific production linked to the use of TMS to treat neuropathic pain and to provide an overview of the research development is comprehensive, well thought out and well motivated.

REPLY: We acknowledge the comment.

2. The presentation/description of Figure 4 is not very informative and could be improved.

REPLY: We improved the presentation/description of Figure 4 (please see the new Figure 4).

3. The question for authors to discuss: What is your opinion about the absence of a dominant source in this topic?

REPLY: We proceeded as suggested (please see lines 310-314).

Reviewer 2 Report

Comments and Suggestions for Authors

This work is well organized and structured.

The authors have identified and mentioned the limitations of this review. I appreciate that they specified these limiting aspects

It must be specified if the authors tried to mitigate errors, if they carefully clean the bibliometric data that they acquire, which includes removing duplicates and erroneous entries.

In order to increase the quality of the analysis, the metrics parameters must be specified in order to enrich the assessment of bibliometric analysis.

Author Response

We acknowledge the comments of the reviewers. These comments were a remarkable contribute to improve the manuscript. A detailed reply to all the comments was included bellow and the manuscript was changed accordingly, as detailed bellow.

Reviewer 2

1. This work is well organized and structured.

REPLY: We acknowledge the comment.

2. The authors have identified and mentioned the limitations of this review. I appreciate that they specified these limiting aspects.

REPLY: We acknowledge the comment.

3. It must be specified if the authors tried to mitigate errors, if they carefully clean the bibliometric data that they acquire, which includes removing duplicates and erroneous entries.

REPLY: We proceeded as suggested (please see lines 136-137). Also, in the section “Top-cited publications” we highlighted only the 3 most-cited “original articles” and the 3 most-cited “review articles” directly related to the primary subject of this study also to minimize possible errors.

4. In order to increase the quality of the analysis, the metrics parameters must be specified in order to enrich the assessment of bibliometric analysis.

REPLY: We proceeded as suggested: we specified the meaning of journal impact factor and field-weighted citation impact in the section 2.2 (please see lines 107-112).

Reviewer 3 Report

Comments and Suggestions for Authors

1) This paper would be given the contribution of TMS to treat Neuropathic pain. It is good to do general overview through biometric analysis for 474 articles. This general overview would be making a contribution to understanding the trends and outline of TMS researches for treating neuropathic pain that have been conducted so far. However, it is still sorry that it is not extended to systematic review or meta analysis that is mentioned as further suggestion in conclusion or some weakness of scientific basis for TMS to treat neuropathic pain

 2) Only publications in Scopus have been considered here. It is recommended that the reason of the other databases has not been considered should be described more in detail even it is decribed in 2.2.

 3) In Figure 2,

- is there any reason in horizontal line, 2023 is not included?    Is there no finding even all publications to July, 2023 were included?

- It is suggested to express the number of some significant year or each year. 

3) In Figure 5, It is suggested to express the number of each country.

Thank you. 

Author Response

We acknowledge the comments of the reviewers. These comments were a remarkable contribute to improve the manuscript. A detailed reply to all the comments was included bellow and the manuscript was changed accordingly, as detailed bellow.

Reviewer 3

1. This paper would be given the contribution of TMS to treat Neuropathic pain. It is good to do general overview through biometric analysis for 474 articles. This general overview would be making a contribution to understanding the trends and outline of TMS researches for treating neuropathic pain that have been conducted so far. However, it is still sorry that it is not extended to systematic review or meta analysis that is mentioned as further suggestion in conclusion or some weakness of scientific basis for TMS to treat neuropathic pain.

REPLY: We acknowledge the comment. In fact, at the present moment the authors do not want to perform a systematic review or meta-analysis of the literature, but we use the bibliometric analysis to set the grounds and identify the scientific production linked to the use of TMS to treat neuropathic pain (our primary objective). After the analysis of the results, in the future we probably will perform a systematic review with meta-analysis but with more precise and defined purposes that the current approach.

2. Only publications in Scopus have been considered here. It is recommended that the reason of the other databases has not been considered should be described more in detail even it is described in 2.2.

REPLY: We proceeded as suggested (please see lines 105-107).

3. In Figure 2,

- is there any reason in horizontal line, 2023 is not included? Is there no finding even all publications to July, 2023 were included?

REPLY: We changed Figure 2 accordingly with a previous reviewer, because the data for 2023 only covers half of the year and could erroneously lead one to believe that the number of publications in the field is significantly decreasing during this year; we added this reason to the manuscript and we added in the text the number of publications until July 6, 2023 (please see lines 176-179).

- it is suggested to express the number of some significant year or each year.

REPLY: We proceeded as suggested (please see the new Figure 2).

4. In Figure 5, It is suggested to express the number of each country.

REPLY: We proceeded as suggested (please see the new Figure 5).

Reviewer 4 Report

Comments and Suggestions for Authors

In the manuscript by Carneiro and Tavares "Transcranial Magnetic Stimulation to Treat Neuropathic Pain: A Bibliometric Analysis". The authors have done a bibliometric analysis for the treatment of neuropathic pain with respect to Transcranial Magnetic Stimulation. I have the following suggestion.

1. I would recommend searching with the type of neuropathic pain as Diabetes, cancer, and others for the analysis. 

Author Response

We acknowledge the comments of the reviewers. These comments were a remarkable contribute to improve the manuscript. A detailed reply to all the comments was included bellow and the manuscript was changed accordingly, as detailed bellow.

Reviewer 4

1. In the manuscript by Carneiro and Tavares "Transcranial Magnetic Stimulation to Treat Neuropathic Pain: A Bibliometric Analysis". The authors have done a bibliometric analysis for the treatment of neuropathic pain with respect to Transcranial Magnetic Stimulation. I have the following suggestion.

I would recommend searching with the type of neuropathic pain as Diabetes, cancer, and others for the analysis. 

REPLY: We acknowledge the comment. We added a new subsection (3.8) in the manuscript dealing with this issue (please see lines 261-273). We also highlighted this in the abstract (please see lines 24-26), in the methods (please see line 126) and in the discussion (please see lines 323-329).